# Theoretical Analysis of Adversarial Learning: A Minimax Approach

**Zhuozhuo Tu[1], Jingwei Zhang[2,1], Dacheng Tao[1]**
[1]UBTECH Sydney AI Centre, School of Computer Science, The University of Sydney, Australia
[2]Department of Computer Science and Engineering, HKUST, Hong Kong
zhtu3055@uni.sydney.edu.au, jzhangey@cse.ust.hk, dacheng.tao@sydney.edu.au

## Abstract

In this paper, we propose a general theoretical method for analyzing the risk bound in the presence of adversaries. Specifically, we try to fit the adversarial learning problem into the minimax framework. We first show that the original adversarial learning problem can be transformed into a minimax statistical learning problem by introducing a transport map between distributions. Then, we prove a new risk bound for this minimax problem in terms of covering numbers under a weak version of Lipschitz condition. Our method can be applied to multi-class classification and popular loss functions including the hinge loss and ramp loss. As some illustrative examples, we derive the adversarial risk bounds for SVMs and deep neural networks, and our bounds have two data-dependent terms, which can be optimized for achieving adversarial robustness.

## 1 Introduction

Machine learning models, especially deep neural networks, have achieved impressive performance across a variety of domains including image classification, natural language processing, and speech recognition. However, these techniques can easily be fooled by adversarial examples, i.e., carefully perturbed input samples aimed to cause misclassification during the test phase. This phenomenon was first studied in spam filtering [14, 31, 32] and has attracted considerable attention since 2014, when Szegedy et al. [42] noticed that small perturbations in images can cause misclassification in neural network classifiers. Since then, there has been considerable focus on developing adversarial attacks against machine learning algorithms [21, 9, 8, 4, 44], and, in response, many defense mechanisms have also been proposed to counter these attacks [22, 20, 15, 41, 33]. These works focus on creating optimization-based robust algorithms, but their generalization performance under adversarial input perturbations is still not fully understood.

Schmidt et al. [38] recently discussed the generalization problem in the adversarial setting and showed that the sample complexity of learning a specific distribution in the presence of $l_\infty$-bounded adversaries increases by an order of $\sqrt{d}$ for all classifiers. The same paper recognized that deriving the agnostic-distribution generalization bound remained an open problem [38]. In a subsequent study, Cullina et al. [13] extended the standard PAC-learning framework to the adversarial setting by defining a corrupted hypothesis class and showed that the VC dimension of this corrupted hypothesis class for halfspace classifiers which controlled the sample complexity does not increase in the presence of an adversary. While their work provided a theoretical understanding of the problem of learning with adversaries, it had two limitations. First, their results could only be applied to binary problems, whereas in practice we usually need to handle multi-class problems. Second, the 0-1 loss function used in their work is not convex and thus very hard to optimize.

In this paper, we propose a general theoretical method for analyzing generalization performance in the presence of adversaries. In particular, we fit the adversarial learning problem into the minimax

framework [28]. In contrast to traditional statistical learning, where the underlying data distribution $P$ is unknown but fixed, the minimax framework considers the uncertainty about the distribution $P$ by introducing an ambiguity set and then aims to minimize the risk with respect to the worst-case distribution in this set. Motivated by Lee & Raginsky [28], we first note that the adversarial expected risk over a distribution $P$ is equivalent to the standard expected risk under a new distribution $P'$. Since this new distribution is not fixed and depends on the hypothesis, we instead consider the worst case. In this way, the original adversarial learning problem is reduced to a minimax problem, and we use the minimax approach to derive the risk bound for the adversarial expected risk. Our contributions can be summarized as follows.

- We propose a general method for analyzing the risk bound in the presence of adversaries. Our method is general in several respects. First, the adversary we consider is general and encompasses all $l_q$ bounded adversaries. Second, our method can be applied to multi-class problems and commonly used loss functions such as the hinge loss and ramp loss, whereas Cullina et al. [13] only considered the binary classification problem and the 0-1 loss.

- We prove a new bound for the local worst-case risk under a weak version of Lipschitz condition. Our bound is always better than that of Lee & Raginsky [29], and can recover the usual risk bound by setting the radius $\epsilon_\mathcal{B}$ of the Wasserstein ball to 0, whereas they give a $\epsilon_\mathcal{B}$-free bound.

- We derive the adversarial risk bounds for SVMs and deep neural networks. Our bounds have two data-dependent terms, suggesting that minimizing the sum of the two terms can help achieve adversarial robustness.

The remainder of this paper is structured as follows. In Section 2, we discuss related works. Section 3 formally defines the problem, and we present our theoretical method in Section 4. The adversarial risk bounds for SVMs and neural networks are described in Section 5, and we conclude and discuss future directions in Section 6.

## 2 Related work

Our work leverages some of the benefits of statistical machine learning, summarized as follows.

### 2.1 Generalization in supervised learning

Generalization is a central problem in supervised learning, and the generalization capability of learning algorithms has been extensively studied. Here we review the salient aspects of generalization in supervised learning relevant to this work.

Two main approaches are used to analyze the generalization bound of a learning algorithm. The first is based on the complexity of the hypothesis class, such as the VC dimension [45, 46] for binary classification, Rademacher and Gaussian complexities [7, 5], and the covering number [53, 52, 6]. Note that hypothesis complexity-based analyses of generalization error are algorithm independent and consider the worst-case generalization over all functions in the hypothesis class. In contrast, the second approach is based on the properties of a learning algorithm and is therefore algorithm dependent. The properties characterizing the generalization of a learning algorithm include, for example, algorithmic stability [11, 39, 30], robustness [50], and algorithmic luckiness [24]. Some other methods exist for analyzing the generalization error in machine learning such as the PAC-Bayesian approach [35, 2], compression-based bounds [27, 3], and information-theoretic approaches [49, 1, 37].

### 2.2 Minimax statistical learning

In contrast to standard empirical risk minimization in supervised learning, where test data follow the same distribution as training data, minimax statistical learning arises in problems of distributionally robust learning [16, 18, 28, 29, 40] and minimizes the worst-case risk over a family of probability distributions. Thus, it can be applied to the learning setting in which the test data distribution differs from that of the training data, such as in domain adaptation and transfer learning [12]. In particular, Gao & Kleywegt [18] proposed a dual representation of worst-case risk over the ambiguity set of probability distributions, which was given by balls in Wasserstein space. Then, Lee & Raginsky

[28] derived the risk bound for minimax learning by exploiting the dual representation of worst-case risk. However, their minimax risk bound would go to infinity and thus become vacuous as $\epsilon_\mathcal{B} \to 0$. Despite that the same authors later presented a new bound [29] by imposing a Lipschitz assumption to avoid this problem, their new bound was $\epsilon_\mathcal{B}$-free and cannot recover the usual risk bound by setting $\epsilon_\mathcal{B} = 0$. Sinha et al. [40] also provided a similar upper bound on the worst-case population loss over distributions defined by the Wasserstein metric via a Lagrangian penalty formulation, and their bound was efficiently computable by a principled adversarial training procedure, which provably certified distributional robustness. However their training procedure required that the penalty parameter should be large enough and thus can only achieve a small amount of robustness. Here we improve on the results in Lee & Raginsky [28, 29] and present a new risk bound for the minimax problem.

## 2.3 Learning with adversaries

The existence of adversaries during the test phase of a learning algorithm may render predictions made by learning system unthrustworthy. There is extensive literature on analysis of adversarial robustness [47, 17, 23, 19] and design of provable defense against adversarial attacks[48, 36, 33, 40], in contrast to the relatively limited literature on risk bound analysis of adversarial learning. A comprehensive review of works on adversarial machine learning can be found in Biggio & Roli [10]. Concurrently to our work, Khim & Loh [25] and Yin et al. [51] provided different approaches to deriving adversarial risk bounds. Khim & Loh [25] derived adversarial risk bounds for linear classifiers and neural networks using a method called supremum transform. However, their approach can only be applied to binary classification. Yin et al. [51] gave similar adversarial risk bounds through the lens of Rademacher complexity. Although they provided risk bounds in multi-class setting, their work focused on $l_\infty$ adversarial attacks and was limited to one-hidden layer ReLU neural networks. After the initial preprint of this paper, Khim & Loh [26] extended their method to multi-class setting by considering the binary supremum transform on each component of classifier, which as a result incurred an extra factor of the number of classes in their bound. Instead we used covering number analysis to derive the multi-class bound, which can avoid explicit dependence on this number.

## 3 Problem setup

We consider a standard statistical learning framework. Let $\mathcal{Z} = \mathcal{X} \times \mathcal{Y}$ be a measurable instance space where $\mathcal{X}$ and $\mathcal{Y}$ represent feature and label spaces, respectively. We assume that examples are independently and identically distributed according to some fixed but unknown distribution $P$. The learning problem is then formulated as follows. The learner considers a class $\mathcal{H}$ of hypothesis $h : \mathcal{X} \to \mathcal{Y}'$ where $\mathcal{Y}'$ sometimes differs from $\mathcal{Y}$ and a loss function $l : \mathcal{Y}' \times \mathcal{Y} \to \mathbb{R}_+$. The learner receives $n$ training examples denoted by $S = ((x_1, y_1), (x_2, y_2), \cdots, (x_n, y_n))$ drawn i.i.d. from $P$ and tries to select a hypothesis $h \in \mathcal{H}$ that has a small expected risk. However, in the presence of adversaries, there will be imperceptible perturbations to the input of examples, which are called adversarial examples. Throughout this paper, we assume that the adversarial examples are generated by adversarially choosing an example from neighborhood $N(x) = \{x' : x' - x \in \mathcal{B}\}$ where $\mathcal{B}$ is a nonempty set. Note that the definition of $N(x)$ is very general and encompasses all $l_q$-bounded adversaries. We next give the formal definition of adversarial expected and empirical risk to measure the learner's performance in the presence of adversaries.

**Definition 1.** *(Adversarial Expected Risk). The adversarial expected risk of a hypothesis $h \in \mathcal{H}$ over the distribution $P$ in the presence of an adversary constrained by $\mathcal{B}$ is*

$$R_P(h, \mathcal{B}) = \mathbb{E}_{(x,y) \sim P}[\max_{x' \in N(x)} l(h(x'), y)].$$

If $\mathcal{B}$ is the zero-dimensional space $\{\mathbf{0}\}$, then the adversarial expected risk will reduce to the standard expected risk without an adversary. Since the true distribution is usually unknown, we instead use the empirical distribution to approximate the true distribution, which is equal to $(x_i, y_i)$ with probability $1/n$ for each $i \in \{1, \cdots, n\}$. That gives us the following definition of adversarial empirical risk.

**Definition 2.** *(Adversarial Empirical Risk ). The adversarial empirical risk of $h$ in the presence of an adversary constrained by $\mathcal{B}$ is*

$$R_{P_n}(h, \mathcal{B}) = \frac{1}{n} \sum_{i=1}^{n} \big[\max_{x' \in N(x_i)} l(h(x'), y_i)\big].$$

# 4 Main results

In this section, we present our main results. The trick is to pushforward the original distribution $P$ into a new distribution $P'$ using a transport map $T_h : \mathcal{Z} \to \mathcal{Z}$ satisfying

$$R_P(h, \mathcal{B}) = R_{P'}(h),$$

where $R_{P'}(h) = \mathbb{E}_{(x,y) \sim P'} l(h(x), y)$ is the standard expected risk without the adversary. Therefore, an upper bound on the expected risk over the new distribution leads to an upper bound on the adversarial expected risk.

Note that the new distribution $P'$ is not fixed and depends on the hypothesis $h$. As a result, traditional statistical learning cannot be directly applied. However, note that these new distributions lie within a Wasserstein ball centered on $P$, which we will show in Section 4.2. If we consider the worst case within this Wasserstein ball, then the original adversarial learning problem can be reduced to a minimax problem. We can therefore use the minimax approach to derive the adversarial risk bound. We first introduce the Wasserstein distance and minimax framework.

## 4.1 Wasserstein distance and local worst-case risk

Let $(\mathcal{Z}, d_{\mathcal{Z}})$ be a metric space where $\mathcal{Z} = \mathcal{X} \times \mathcal{Y}$ and $d_{\mathcal{Z}}$ is defined as

$$d_{\mathcal{Z}}^p(z, z') = d_{\mathcal{Z}}^p((x,y),(x',y')) = (d_{\mathcal{X}}^p(x,x') + d_{\mathcal{Y}}^p(y,y'))$$

with $d_{\mathcal{X}}$ and $d_{\mathcal{Y}}$ representing the metric in the feature space and label space respectively. For example, if $\mathcal{Y} = \{1, -1\}$, $d_{\mathcal{Y}}(y, y')$ can be $\mathbb{1}_{(y \neq y')}$, and if $\mathcal{Y} = [-B, B]$, $d_{\mathcal{Y}}(y, y')$ can be $(y - y')^2$. In this paper, we require that $d_{\mathcal{X}}$ is translation invariant, i.e., $d_{\mathcal{X}}(x, x') = d_{\mathcal{X}}(x - x', 0)$. With this metric, we denote with $\mathcal{P}(\mathcal{Z})$ the space of all Borel probability measures on $\mathcal{Z}$, and with $\mathcal{P}_p(\mathcal{Z})$ the space of all $P \in \mathcal{P}(\mathcal{Z})$ with finite $p$th moments for $p \geq 1$:

$$\mathcal{P}_p(\mathcal{Z}) := \{P \in \mathcal{P}(\mathcal{Z}) : \mathbb{E}_P[d_{\mathcal{Z}}^p(z, z_0)] < \infty \ for \ z_0 \in \mathcal{Z}\}.$$

Then, the $p$-Wasserstein distance between two probability measures $P, Q \in \mathcal{P}_p(\mathcal{Z})$ is defined as

$$W_p(P, Q) := \inf_{M \in \Gamma(P,Q)} (\mathbb{E}_{(z,z') \sim M}[d_{\mathcal{Z}}^p(z, z')])^{1/p},$$

where $\Gamma(P, Q)$ denotes the collection of all measures on $\mathcal{Z} \times \mathcal{Z}$ with marginals P and Q on the first and second factors, respectively.

Now we define the local worst-case risk of $h$ at $P$,

$$R_{\epsilon,p}(P, h) := \sup_{Q \in B_{\epsilon,p}^W(P)} R_Q(h),$$

where $B_{\epsilon,p}^W(P) := \{Q \in \mathcal{P}_p(Z) : W_p(P, Q)) \leq \epsilon\}$ is the $p$-Wasserstein ball of radius $\epsilon \geq 0$ centered at $P$.

With these definitions, we next show the adversarial expected risk can be related to the local worst-case risk by a transport map $T_h$.

## 4.2 Transport map

Define a mapping $T_h : \mathcal{Z} \to \mathcal{Z}$

$$z = (x, y) \to (x^*, y),$$

where $x^* = \arg\max_{x' \in N(x)} l(h(x'), y)$. By the definition of $d_{\mathcal{Z}}$, it is easy to obtain $d_{\mathcal{Z}}((x,y),(x^*,y)) = d_{\mathcal{X}}(x, x^*)$. We now prove that the adversarial expected risk can be related to the standard expected risk via the mapping $T_h$.

**Lemma 1.** *Let $P' = T_h \# P$, the pushforward of $P$ by $T_h$, then we have*

$$R_P(h, \mathcal{B}) = R_{P'}(h).$$

*Proof.* By the definition, we have

$$R_P(h, \mathcal{B}) = \mathbb{E}_{(x,y) \sim P}[\max_{x' \in N(x)} l(h(x'), y)] = \mathbb{E}_{(x,y) \sim P}[l(h(x^*), y)] = \mathbb{E}_{(x,y) \sim P'}[l(h(x), y)] .$$

So $R_P(h, \mathcal{B}) = R_{P'}(h)$. $\qquad \square$

By this lemma, the adversarial expected risk over a distribution $P$ is equivalent to the standard expected risk over a new distribution $P'$. However since the new distribution is not fixed and depends on the hypothesis $h$, traditional statistical learning cannot be directly applied. Luckily, the following lemma proves that all these new distributions locate within a Wasserstein ball centered at $P$.

**Lemma 2.** *Define the radius of the adversary constrained by $\mathcal{B}$ as $\epsilon_{\mathcal{B}} := \sup_{x \in \mathcal{B}} d_{\mathcal{X}}(x, 0)$. For any hypothesis $h$ and the corresponding $P' = T_h \# P$, we have*

$$W_p(P, P') \leq \epsilon_{\mathcal{B}}.$$

*Proof.* By the definition of Wasserstein distance,

$$W_p^p(P, P') \leq \mathbb{E}_P[d_{\mathcal{Z}}^p(Z, T_h(Z))] = \mathbb{E}_P[d_{\mathcal{X}}^p(x, x^*)] \leq \epsilon_{\mathcal{B}}^p,$$

where the last inequality uses the translation invariant property of $d_{\mathcal{X}}$. Therefore, we have $W_p(P, P') \leq \epsilon_{\mathcal{B}}$. $\qquad\square$

From this lemma, we can see that all possible new distributions lie within a Wasserstein ball of radius $\epsilon_{\mathcal{B}}$ centered on $P$. So, by upper bounding the worst-case risk in the ball, we can bound the adversarial expected risk. The relationship between local worst-case risk and adversarial expected risk is as follows. Note that this inequality holds for any $p \geq 1$. For ease of exposition, in the rest of the paper, we only discuss the case $p = 1$; that is,

$$R_P(h, \mathcal{B}) \leq R_{\epsilon_{\mathcal{B}}, 1}(P, h), \quad \forall h \in \mathcal{H}. \tag{1}$$

## 4.3 Adversarial risk bounds

In this subsection, we first prove a bound for the local worst-case risk. Then, the adversarial risk bounds can be derived directly by (1). To simplify notation, we denote a function class $\mathcal{F}$ by compositing the functions in $\mathcal{H}$ with the loss function $l(\cdot, \cdot)$, i.e., $\mathcal{F} = \{(x, y) \to l(h(x), y) : h \in \mathcal{H}\}$. The key ingredient of a bound on the local worst-case risk is the following strong duality result by Gao & Kleywegt [18]:

**Proposition 1.** *For any upper semicontinuous function $f : \mathcal{Z} \to \mathbb{R}$ and for any $P \in \mathcal{P}_p(\mathcal{Z})$,*

$$R_{\epsilon_{\mathcal{B}}, 1}(P, f) = \min_{\lambda \geq 0}\{\lambda \epsilon_{\mathcal{B}} + \mathbb{E}_P[\varphi_{\lambda, f}(z)]\},$$

*where $\varphi_{\lambda, f}(z) := \sup_{z' \in \mathcal{Z}}\{f(z') - \lambda \cdot d_{\mathcal{Z}}(z, z')\}$.*

We begin with some assumptions.

**Assumption 1.** *The instance space $\mathcal{Z}$ is bounded: $diam(\mathcal{Z}) := \sup_{z, z' \in \mathcal{Z}} d_{\mathcal{Z}}(z, z') < \infty$.*

**Assumption 2.** *The functions in $\mathcal{F}$ are upper semicontinuous and uniformly bounded: $0 \leq f(z) \leq M < \infty$ for all $f \in \mathcal{F}$ and $z \in \mathcal{Z}$.*

**Assumption 3.** *For any function $f \in \mathcal{F}$ and any $z \in \mathcal{Z}$, there exists $\lambda_{f,z}$ such that $f(z') - f(z) \leq \lambda_{f,z} d_{\mathcal{Z}}(z, z')$ for any $z' \in \mathcal{Z}$.*

Note that Assumption 3 is a weak version of Lipschitz condition since $\lambda_{f,z}$ is not fixed and depends on $f$ and $z$. It is easy to see that if the function $f \in \mathcal{F}$ is $L$-Lipschitz with respect to the metric $d_{\mathcal{Z}}$, i.e., $|f(z) - f(z')| \leq L d_{\mathcal{Z}}(z, z')$, Assumption 3 automatically holds with $\lambda_{f,z}$ always being $L$. Now we give an equivalent expression for Assumption 3 which is easier to use in our proofs.

**Lemma 3.** *Assumption 3 holds if and only if for any function $f \in \mathcal{F}$ and any empirical distribution $P_n$, the set $\{\lambda : \psi_{f,P_n}(\lambda) = 0\}$ is nonempty, where $\psi_{f,P_n}(\lambda) := \mathbb{E}_{P_n}(\sup_{z' \in \mathcal{Z}}\{f(z') - \lambda d_{\mathcal{Z}}(z, z') - f(z)\})$.*

The proof of Lemma 3 is contained in Appendix A.

We denote the smallest value in the set as $\lambda_{f,P_n}^+ := \inf\{\lambda : \psi_{f,P_n}(\lambda) = 0\}$. In order to prove the local worst-case risk bound, we need two technical lemmas.

**Lemma 4.** *Fix some $f \in \mathcal{F}$. Define $\bar{\lambda}$ via*

$$\bar{\lambda} := \arg\min_{\lambda \geq 0}\{\lambda \epsilon_{\mathcal{B}} + \mathbb{E}_{P_n}[\varphi_{\lambda, f}(Z)]\}.$$

*Then*

$$\bar{\lambda} \in \begin{cases} [0, \dfrac{M}{\epsilon_{\mathcal{B}}}] & if \; \epsilon_{\mathcal{B}} \geq \dfrac{M}{\lambda^+_{f,P_n}} \\[2ex] [\lambda^-_{f,P_n}, \lambda^+_{f,P_n}] & if \; \epsilon_{\mathcal{B}} < \dfrac{M}{\lambda^+_{f,P_n}} \end{cases} \quad , \tag{2}$$

*where* $\lambda^-_{f,P_n} := \sup\{\lambda : \psi_{f,P_n}(\lambda) = \lambda^+_{f,P_n} \cdot \epsilon_{\mathcal{B}}\}$ *if the set* $\{\lambda : \psi_{f,P_n}(\lambda) = \lambda^+_{f,P_n} \cdot \epsilon_{\mathcal{B}}\}$ *is nonempty, otherwise* $\lambda^-_{f,P_n} := 0$.

**Remark 1.** We can show that $\lim_{\epsilon_{\mathcal{B}} \to 0} \lambda^-_{f,P_n} = \lambda^+_{f,P_n}$ by using $(\epsilon, \delta)$ language as follows. $\forall \epsilon > 0$, define $\delta = \frac{\psi_{f,P_n}(\lambda^+_{f,P_n} - \epsilon)}{\lambda^+_{f,P_n}}$. Then, for any $\epsilon_{\mathcal{B}} < \delta$, we have $\psi_{f,P_n}(\lambda^+_{f,P_n} - \epsilon) > \lambda^+_{f,P_n} \cdot \epsilon_{\mathcal{B}}$. By the definition of $\lambda^-_{f,P_n}$, $\psi_{f,P_n}(\lambda^-_{f,P_n}) = \lambda^+_{f,P_n} \cdot \epsilon_{\mathcal{B}}$. Since $\psi_{f,P_n}(\lambda)$ is monotonically non-increasing, we have $\lambda^-_{f,P_n} > \lambda^+_{f,P_n} - \epsilon$. Therefore, $\lim_{\epsilon_{\mathcal{B}} \to 0} \lambda^-_{f,P_n} = \lambda^+_{f,P_n}$.

**Lemma 5.** *Define the function class* $\Phi := \{\varphi_{\lambda,f} : \lambda \in [a,b], f \in \mathcal{F}\}$ *where* $b \geq a \geq 0$. *Then, the expected Rademacher complexity of the function class* $\Phi$ *satisfies*

$$\mathfrak{R}_n(\Phi) \leq \frac{12\mathfrak{C}(\mathcal{F})}{\sqrt{n}} + \frac{6\sqrt{\pi}}{\sqrt{n}}(b-a) \cdot diam(Z),$$

*where* $\mathfrak{C}(\mathcal{F}) := \int_0^\infty \sqrt{log\mathcal{N}(\mathcal{F}, ||\cdot||_\infty, u/2)}du$ *and* $\mathcal{N}(\mathcal{F}, ||\cdot||_\infty, u/2)$ *denotes the covering number of* $\mathcal{F}$.

The proofs of Lemma 4 and 5 is contained in Appendix B.

We are now ready to prove the local worst-case risk bound. Let $\bar{\lambda} \in [\zeta^-_{f,P_n}, \zeta^+_{f,P_n}]$ denotes expression (2), $[\zeta^-, \zeta^+] := \bigcup_{f,P_n}[\zeta^-_{f,P_n}, \zeta^+_{f,P_n}]$ and $\Lambda_{\epsilon_{\mathcal{B}}} := \zeta^+ - \zeta^-$. It is straightforward to check that $[\zeta^-, \zeta^+] \subset [0, M/\epsilon_{\mathcal{B}}]$ from expression (2). The generalization bound for local worst-case risk is given by the following lemma.

**Lemma 6.** *If the assumptions 1- 3 hold, then for any* $f \in \mathcal{F}$, *we have*

$$R_{\epsilon_{\mathcal{B}},1}(P,f) - R_{\epsilon_{\mathcal{B}},1}(P_n,f) \leq \frac{24\mathfrak{C}(\mathcal{F})}{\sqrt{n}} + M\sqrt{\frac{log(\frac{1}{\delta})}{2n}} + \frac{12\sqrt{\pi}}{\sqrt{n}}\Lambda_{\epsilon_{\mathcal{B}}} \cdot diam(Z)$$

*with probability at least* $1 - \delta$.

**Remark 2.** Lee & Raginsky [29] proved a bound with $\Lambda_{\epsilon_{\mathcal{B}}} \equiv L$ under the Lipschitz assumption where $L$ represents the Lipschitz constant. Our result improves a lot on theirs. First, our Assumption 3 is weaker than their Lipschitz assumption. Second, even under the weaker assumptions, our bound is always better than their results since $[\zeta^-, \zeta^+] \subset [0, L]$ by expression (2) and the definition of $\lambda^+_{f,P_n}$. Finally, by setting $\epsilon_B = 0$, the term $\frac{12\sqrt{\pi}}{\sqrt{n}}\Lambda_{\epsilon_{\mathcal{B}}} \cdot diam(Z)$ in our bound will vanish, recovering the usual risk bound, whereas they gave a $\epsilon_{\mathcal{B}}$-free bound with $\Lambda_{\epsilon_{\mathcal{B}}}$ always being the constant $L$.

This leads to our main theorem for the adversarial expected risk.

**Theorem 1.** *If the assumptions 1- 3 hold, for any* $f \in \mathcal{F}$, *we have*

$$R_P(f, \mathcal{B}) \leq \frac{1}{n}\sum_{i=1}^n f(z_i) + \min_{\lambda \geq 0}\{\lambda\epsilon_{\mathcal{B}} + \psi_{f,P_n}(\lambda)\} + \frac{24\mathfrak{C}(\mathcal{F})}{\sqrt{n}} + \frac{12\sqrt{\pi}}{\sqrt{n}}\Lambda_{\epsilon_{\mathcal{B}}} \cdot diam(Z) + M\sqrt{\frac{log(\frac{1}{\delta})}{2n}} \tag{3}$$

*and*

$$R_P(f, \mathcal{B}) \leq \frac{1}{n}\sum_{i=1}^n f(z_i) + \lambda^+_{f,P_n}\epsilon_{\mathcal{B}} + \frac{24\mathfrak{C}(\mathcal{F})}{\sqrt{n}} + \frac{12\sqrt{\pi}}{\sqrt{n}}\Lambda_{\epsilon_{\mathcal{B}}} \cdot diam(Z) + M\sqrt{\frac{log(\frac{1}{\delta})}{2n}} \tag{4}$$

*with probability at least* $1 - \delta$.

**Remark 3.** We are interested in how the adversarial risk bounds differ from the case in which the adversary is absent. Plugging $\epsilon_\mathcal{B} = 0$ into inequality (3) or (4) yields the usual generalization bound of the form

$$R_P(h) \leq \frac{1}{n}\sum_{i=1}^n f(z_i) + \frac{24\mathfrak{C}(\mathcal{F})}{\sqrt{n}} + M\sqrt{\frac{log(1/\delta)}{2n}}.$$

So the effect of an adversary is to introduce an extra complexity term $12\sqrt{\pi}\Lambda_{\epsilon_\mathcal{B}} \cdot diam(Z)/\sqrt{n}$ and an additional term on $\epsilon_\mathcal{B}$ which contributes to the empirical risk.

**Remark 4.** The extra complexity term will decrease as $\epsilon_\mathcal{B}$ gets bigger if $\epsilon_\mathcal{B} \geq M/\lambda_{f,P_n}^+$ by expression (2), indicating that a stronger adversary might have a negative impact on the hypothesis class complexity. This is intuitive, since different hypotheses might have the same performance in the presence of a strong adversary and, therefore, the hypothesis class complexity will decrease. We emphasize that this phenomenon does not occur in concurrent works [25, 51]. In both of their work, this term will increase linearly as $\epsilon_\mathcal{B}$ grows.

**Remark 5.** There are two data dependent terms $1/n\sum_{i=1}^n f(z_i)$ and $\min_{\lambda \geq 0}\{\lambda\epsilon_\mathcal{B} + \psi_{f,P_n}(\lambda)\}$ (or $\lambda_{f,P_n}^+ \epsilon_\mathcal{B}$) in bound (3) (or (4)), corresponding to the empirical risk and the effect of adversary on empirical risk, respectively. Although the bound (3) is tighter, it is hard to optimize because of the inner minimization problem. The bound (4) cannot be directly minimized either because $\lambda_{f,P_n}^+$ is computationally intractable in practice. But we can consider an upper bound for $\lambda_{f,P_n}^+$. For example, if $f$ is $L$-Lipschitz, by the definition of $\lambda_{f,P_n}^+$, we have $\lambda_{f,P_n}^+ \leq L$. See Section 5 for more examples. This upper bound for $\lambda_{f,P_n}^+$ can be used in optimization, as we will discuss in Section 6. In particular, if $\psi_{f,P_n}(\lambda) \equiv 0$ for any $\lambda \geq 0$, we get $\lambda_{f,P_n}^+ = 0$, and the additional term $\lambda_{f,P_n}^+ \epsilon_\mathcal{B}$ in inequality (4) will disappear.

# 5 Example bounds

In this section, we illustrate the application of Theorem 1 to two commonly-used models: SVMs and neural networks.

## 5.1 Support vector machines

We first start with SVMs. Let $\mathcal{Z} = \mathcal{X} \times \mathcal{Y}$, where the feature space $\mathcal{X} = \{x \in \mathbb{R}^d : ||x||_2 \leq r\}$ and the label space $\mathcal{Y} = \{-1, +1\}$. Equip $\mathcal{Z}$ with the Euclidean metric

$$d_\mathcal{Z}(z, z') = d_\mathcal{Z}((x, y), (x', y')) = ||x - x'||_2 + \mathbb{1}_{(y \neq y')}.$$

Consider the hypothesis space $\mathcal{F} = \{(x, y) \to \max\{0, 1 - yh(x)\} : h \in H\}$, where $H = \{x \to w \cdot x : ||w||_2 \leq \Lambda\}$. We can now derive the expected risk bound for SVMs in the presence of an adversary.

**Corollary 1.** *For the SVMs setting considered above, for any $f \in \mathcal{F}$, with probability at least $1 - \delta$,*

$$R_P(f, \mathcal{B}) \leq \frac{1}{n}\sum_{i=1}^n f(z_i) + \lambda_{f,P_n}^+ \epsilon_\mathcal{B} + \frac{144}{\sqrt{n}}\Lambda r\sqrt{d} + \frac{12\sqrt{\pi}}{\sqrt{n}}\Lambda_{\epsilon_\mathcal{B}} \cdot (2r + 1) + (1 + \Lambda r)\sqrt{\frac{log(\frac{1}{\delta})}{2n}},$$

*where $\lambda_{f,P_n}^+ \leq \max_i\{2y_i w \cdot x_i, ||w||_2\}$.*

The proof of Corollary 1 can be found in Appendix E.

## 5.2 Neural networks

We next consider feed-forward neural networks. To demonstrate the generality of our method, we consider a multi-class prediction problem. Let $\mathcal{Z} = \mathcal{X} \times \mathcal{Y}$, where the feature space $\mathcal{X} = \{x \in \mathbb{R}^d : ||x||_2 \leq B\}$ and the label space $\mathcal{Y} = \{1, 2, \cdots, k\}$; $k$ represents the number of classes. The network uses $L$ fixed nonlinear activation functions $(\sigma_1, \sigma_2, \cdots, \sigma_L)$, where $\sigma_i$ is $\rho_i$-Lipschitz and satisfies $\sigma_i(0) = 0$. Given $L$ weight matrices $\mathcal{A} = (A_1, A_2, \cdots, A_L)$, the network computes the following function

$$\mathcal{H}_\mathcal{A}(x) := \sigma_L(A_L\sigma_{L-1}(A_{L-1}\sigma_{L-2}(\cdots\sigma_2(A_2\sigma_1(A_1x)\cdot)),$$

where $A_i \in \mathbb{R}^{d_i \times d_{i-1}}$ and $\mathcal{H}_{\mathcal{A}} : \mathbb{R}^d \to \mathbb{R}^k$ with $d_0 = d$ and $d_L = k$. Let $W = \max\{d_0, d_1, \cdots, d_L\}$. Define a margin operator $\mathcal{M} : \mathbb{R}^k \times \{1, 2, \cdots, k\} \to \mathbb{R}$ as $\mathcal{M}(v, y) := v_y - \max_{j \neq y} v_j$ and the ramp loss $l_\gamma : \mathbb{R} \to \mathbb{R}^+$ as

$$l_\gamma := \begin{cases} 0 & r < -\gamma \\ 1 + r/\gamma & r \in [-\gamma, 0] \\ 1 & r > 0 \end{cases} .$$

Consider the hypothesis class $\mathcal{F} = \{(x, y) \to l_\gamma(-\mathcal{M}(\mathcal{H}_{\mathcal{A}}(x), y)) : \mathcal{A} = (A_1, A_2, \cdots, A_L), \|A_i\|_\sigma \leq s_i, \|A_i\|_F \leq b_i\}$, where $\|\cdot\|_\sigma$ represents spectral norm and $\|\cdot\|_F$ denotes the Frobenius norm. The metric in space $\mathcal{Z}$ is defined as

$$d_{\mathcal{Z}}(z, z') = d_{\mathcal{Z}}((x, y), (x', y')) = \|x - x'\|_2 + \mathbb{1}_{(y \neq y')}.$$

Now we derive the adversarial expected risk for neural networks.

**Corollary 2.** *For the neural networks setting defined above, for any $f \in \mathcal{F}$, with probability of $1 - \delta$, the following inequality holds*

$$R_P(f, \mathcal{B}) \leq \frac{1}{n} \sum_{i=1}^n f(z_i) + \lambda^+_{f,P_n} \epsilon_{\mathcal{B}} + \frac{288}{\gamma \sqrt{n}} \prod_{i=1}^L \rho_i s_i BW \left( \sum_{i=1}^L \left( \frac{b_i}{s_i} \right)^{\frac{1}{2}} \right)^2 +$$

$$\frac{12\sqrt{\pi}}{\sqrt{n}} \Lambda_{\epsilon_{\mathcal{B}}} \cdot (2B + 1) + \sqrt{\frac{log(1/\delta)}{2n}}$$

*where* $\lambda^+_{f,P_n} \leq \max_j \left\{ \frac{2}{\gamma} \prod_{i=1}^L \rho_i \|A_i\|_\sigma, \frac{1}{\gamma} \left( \mathcal{M}(\mathcal{H}_{\mathcal{A}}(x_j), y_j) + \max \mathcal{H}_{\mathcal{A}}(x_j) - \min \mathcal{H}_{\mathcal{A}}(x_j) \right) \right\}.$

The proof of this Corollary is provided in Appendix F.

**Remark 6.** Setting $\epsilon_{\mathcal{B}} = 0$, we obtain a risk bound for neural networks:

$$R_P(f) \leq \frac{1}{n} \sum_{i=1}^n f(z_i) + \sqrt{\frac{log(1/\delta)}{2n}} + \frac{288}{\gamma \sqrt{n}} \prod_{i=1}^L \rho_i s_i BW \left( \sum_{i=1}^L \left( \frac{b_i}{s_i} \right)^{1/2} \right)^2. \quad (5)$$

The bound is in terms of the spectral norm and the Frobenius norm. Although inequality (5) is similar to the results in Bartlett et al. [6] and Neyshabur et al. [35], since our proof technique is different, our approach may provide a different perspective on the generalization of deep neural networks.

## 6   Conclusions

In this paper, we propose a theoretical method for deriving adversarial risk bounds. Our method is general and can easily be applied to multi-class problems and most of the commonly used loss functions. The bound may be loose in some cases, since we consider the worst case distribution in the Wasserstein ball to avoid computing the transport map. However, for some problems, it may be possible to derive the transport map and thus get tighter bounds. Furthermore, our bounds may be made tighter by relying on the expected Rademacher complexity directly instead of using covering numbers.

In the future, one interesting problem is to develop adversarial robust algorithms based on our results. For example, our bounds suggest that minimizing the sum of empirical risk and the term $\lambda^+_{f,P_n} \epsilon_{\mathcal{B}}$ can help achieve adversarial robustness. However, since $\lambda^+_{f,P_n}$ is computationally intractable in practice, instead of using the exact $\lambda^+_{f,P_n}$ in the objective function, we may consider the data-dependent upper bound for $\lambda^+_{f,P_n}$ which is usually easier to obtain and a regularization parameter $\eta \in [0, 1]$ selected via grid search. For a fixed $\eta$, we multiply it by the upper bound for $\lambda^+_{f,P_n}$ and use this product as a surrogate of the true $\lambda^+_{f,P_n}$ in the objective function. Afterward, we minimize this surrogate objective function and obtain the optimal solution for this specific $\eta$. Each such $\eta$ corresponds to a solution. Finally we choose the best one from these candidates.

### Acknowledgments

We thank the reviewers for their constructive comments that helped improve the paper significantly. This work was supported by the ARC FL-170100117.

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
