[Supplementary Material · 6644-Supplementary Material.pdf]

# Supplementary Material

## A   Proof of Lemma 3

*Proof of Lemma 3.* We first prove necessity. For any $f \in \mathcal{F}$ and any empirical distribution $z_i \sim P_n$, by Assumption 3, there exists $\lambda_{f,z_i}$ such that $f(z') - f(z_i) \leq \lambda_{f,z_i} d_{\mathcal{Z}}(z_i, z')$ for any $z' \in \mathcal{Z}$, which leads to $\sup_{z' \in \mathcal{Z}} \{f(z') - \lambda_{f,z_i} d_{\mathcal{Z}}(z_i, z') - f(z_i)\} = 0$. Let $\lambda^* = \max_i \{\lambda_{z_i,f}\}$. Then, for any $z_i \sim P_n$, we have $\sup_{z' \in \mathcal{Z}} \{f(z') - \lambda^* d_{\mathcal{Z}}(z_i, z') - f(z_i)\} = 0$. Therefore, $\psi_{f,P_n}(\lambda^*) = 0$ and the set $\{\lambda : \psi_{f,P_n}(\lambda) = 0\}$ is nonempty. The sufficiency is obvious by the definition of $\psi_{f,P_n}(\lambda)$.

## B   Proofs of Lemma 4 and 5

*Proof of Lemma 4.* If $\epsilon_{\mathcal{B}} \geq \dfrac{M}{\lambda_{f,P_n}^+}$, by Proposition 1, $R_{\epsilon_{\mathcal{B}},1}(P_n, f) = \bar{\lambda} \epsilon_{\mathcal{B}} + \mathbb{E}_{P_n}[\varphi_{\bar{\lambda},f}(Z)]$, we have

$$\bar{\lambda} \epsilon_{\mathcal{B}} \leq R_{\epsilon_{\mathcal{B}},1}(P_n, f).$$

Since $f(z) \leq M$ for any $z$, we get $R_{\epsilon_{\mathcal{B}},1}(P_n, f) \leq M$. So $\bar{\lambda} \leq \dfrac{M}{\epsilon_{\mathcal{B}}}$.

For the other side, we first show that $\psi_{f,P_n}(\lambda)$ is continuous and monotonically non-increasing. The monotonicity is easy to verify from the definition. For continuity, for any $\lambda_2 > \lambda_1$, suppose that

$$\hat{z} = \sup_{z' \in \mathcal{Z}} \{f(z') - \lambda_1 d_{\mathcal{Z}}(z, z') - f(z)\}),$$
$$z^* = \sup_{z' \in \mathcal{Z}} \{f(z') - \lambda_2 d_{\mathcal{Z}}(z, z') - f(z)\}).$$

Then we have

$$\psi_{f,P_n}(\lambda_1) - \psi_{f,P_n}(\lambda_2)$$
$$= \mathbb{E}_{P_n}(\sup_{z' \in \mathcal{Z}} \{f(z') - \lambda_1 d_{\mathcal{Z}}(z, z') - f(z)\} - \sup_{z' \in \mathcal{Z}} \{f(z') - \lambda_2 d_{\mathcal{Z}}(z, z') - f(z)\}) .$$
$$\leq \mathbb{E}_{p_n}((\lambda_2 - \lambda_1) d_{\mathcal{Z}}(z, \hat{z})) \leq (\lambda_2 - \lambda_1) diam(\mathcal{Z})$$

So $\psi_{f,P_n}(\lambda)$ is $diam(\mathcal{Z})$-Lipschitz and thus continuous.

Now we prove $\bar{\lambda} \in [\lambda_{f,P_n}^-, \lambda_{f,P_n}^+]$. If $\lambda > \lambda_{f,P_n}^+$, by the monotonicity and nonnegativity of $\psi_{f,P_n}(\lambda)$, we have $\psi_{f,P_n}(\lambda) = \psi_{f,P_n}(\lambda_{f,P_n}^+) = 0$, which implies $\lambda \epsilon_{\mathcal{B}} + \mathbb{E}_{P_n}[\varphi_{\lambda,f}(z)] \geq \lambda_{f,P_n}^+ \epsilon_{\mathcal{B}} + \mathbb{E}_{P_n}[\varphi_{\lambda_{f,P_n}^+,f}(z)]$. Therefore the optimal $\bar{\lambda} \leq \lambda_{f,P_n}^+$. To show $\bar{\lambda} \geq \lambda_{f,P_n}^-$, first notice that $\psi_{f,P_n}(\lambda)$ belongs to $[0, M]$ for any $\lambda$. We define

$$\lambda_{f,P_n}^- := \sup\{\lambda : \psi_{f,P_n}(\lambda) = \lambda_{f,P_n}^+ \cdot \epsilon_{\mathcal{B}}\}.$$

Note that this set $\{\lambda : \psi_{f,P_n}(\lambda) = \lambda_{f,P_n}^+ \cdot \epsilon_{\mathcal{B}}\}$ might be empty if $\psi_{f,P_n}(0) < \lambda_{f,P_n}^+ \cdot \epsilon_{\mathcal{B}} < M$. In this case, we just let $\lambda_{f,P_n}^- = 0$, and $\bar{\lambda}$ must belong to $[0, \lambda_{f,P_n}^+]$. Otherwise, there must exist some $\lambda \in [0, \lambda_{f,P_n}^+]$ which satisfies $\psi_{f,P_n}(\lambda) = \lambda_{f,P_n}^+ \cdot \epsilon_{\mathcal{B}}$ by the intermediate value theorem of a continuous function. We choose $\lambda_{f,P_n}^-$ to be the maximal one in that set. Then, for any $\lambda < \lambda_{f,P_n}^-$, since $\psi_{f,P_n}(\lambda)$ is monotonically non-increasing, we have

$$\mathbb{E}_{P_n}(\sup_{z' \in \mathcal{Z}} \{f(z') - \lambda d_{\mathcal{Z}}(z, z') - f(z)\}) \geq \lambda_{f,P_n}^+ \cdot \epsilon_{\mathcal{B}}.$$

By rearranging the items on both sides, we obtain

$$\lambda \epsilon_{\mathcal{B}} + \mathbb{E}_{P_n}[\varphi_{\lambda,f}(z)] \geq \lambda_{f,P_n}^+ \cdot \epsilon_{\mathcal{B}} + \mathbb{E}_{P_n}(f(z))$$

for any $\lambda < \lambda_{f,P_n}^-$. Therefore, $\bar{\lambda} \geq \lambda_{f,P_n}^-$, and we complete the proof.

*Proof of Lemma 5.* Define the $\Phi$-indexed process $X = (X_\varphi)_{\varphi \in \Phi}$ by

$$X_\varphi := \frac{1}{\sqrt{n}} \sum_{i=1}^{n} \sigma_i \varphi(z_i).$$

Note that $\mathbb{E}[X_\varphi] = 0$ for all $\varphi \in \Phi$. First we show that X is a subgaussian process with respect to the pseudometric $d_\Phi(\varphi, \varphi')$, defined as

$$d_\Phi(\varphi, \varphi') := ||f - f'||_\infty + diam(Z) \cdot |\lambda - \lambda'|$$

for $\varphi = \varphi_{\lambda, f}$ and $\varphi' = \varphi_{\lambda', f'}$. From the definition of $\varphi_{\lambda, f}$, it is easy to show that $||\varphi - \varphi'||_\infty \leq d_\Phi(\varphi, \varphi')$. Then for any $t \in \mathbb{R}$, we can get

$$\mathbb{E}\left[\exp(t(X_\varphi - X'_\varphi))\right] = \mathbb{E}\left[\exp(\frac{t}{\sqrt{n}} \sum_{i=1}^n \sigma_i(\varphi(z_i) - \varphi'(z_i)))\right]$$
$$= \left(\mathbb{E}\left[\exp(\frac{t}{\sqrt{n}} \sigma_1(\varphi(z_1) - \varphi'(z_1)))\right]\right)^n \leq \exp\left(\frac{t^2 d_\Phi^2(\varphi, \varphi')}{2}\right) ,$$

where the second equality is by the fact that $(\sigma_i, z_i)$ are i.i.d., and the final inequality uses Hoeffding's lemma. Therefore, X is subgaussian with respect to $d_\Phi$. And the expected Rademacher complexity $\mathfrak{R}_n(\Phi)$ can be bounded by the Dudley entropy integral [43]:

$$\mathfrak{R}_n(\Phi) \leq \frac{12}{\sqrt{n}} \int_0^\infty \sqrt{log\mathcal{N}(\Phi, d_\Phi, u)} du,$$

where $\mathcal{N}(\Phi, d_\Phi, \cdot)$ represents the covering numbers of $(\Phi, d_\Phi)$. By the definition of $d_\Phi$, it follows that

$$\mathcal{N}(\Phi, d_\Phi, u) \leq \mathcal{N}(\mathcal{F}, || \cdot ||_\infty, u/2) \cdot \mathcal{N}([a, b], | \cdot |, \frac{u}{2 \cdot diam(Z)})$$

and therefore

$$\mathfrak{R}_n(\Phi) \leq \frac{12}{\sqrt{n}} \left(\int_0^\infty \sqrt{log\mathcal{N}(\mathcal{F}, || \cdot ||_\infty, u/2)} du + \int_0^\infty \sqrt{log\mathcal{N}([a, b], | \cdot |, u/(2 \cdot diam(Z)))} du\right) .$$

The second integral term could be easily obtained as follows

$$\int_0^\infty \sqrt{log\mathcal{N}([a, b], | \cdot |, u/(2 \cdot diam(Z)))} du \leq (b - a) \cdot diam(z) \int_0^1 \sqrt{log\frac{1}{u}} du = \frac{\sqrt{\pi}}{2}(b - a) \cdot diam(z) .$$

Consequently,

$$\mathfrak{R}_n(\Phi) \leq \frac{12}{\sqrt{n}} \int_0^\infty \sqrt{log\mathcal{N}(\mathcal{F}, || \cdot ||_\infty, u/2)} du + \frac{6\sqrt{\pi}}{\sqrt{n}}(b - a) \cdot diam(Z) .$$

## C   Proof of Lemma 6

*Proof of Lemma 6.* For any $f \in \mathcal{F}$, define

$$\bar{\lambda} := \arg\min_{\lambda \geq 0}\{\lambda \epsilon_\mathcal{B} + \mathbb{E}_{P_n}[\varphi_{\lambda, f}(Z)]\}.$$

Then using Proposition 1, we can write

$$R_{\epsilon_\mathcal{B}, 1}(P, f) - R_{\epsilon_\mathcal{B}, 1}(P_n, f)$$
$$= \min_{\lambda \geq 0}\left\{\lambda \epsilon_\mathcal{B} + \int_\mathcal{Z} \varphi_{\lambda, f}(z)P(dz)\right\} - \left(\bar{\lambda}\epsilon_\mathcal{B} + \int_\mathcal{Z} \varphi_{\bar{\lambda}, f}(z)P_n(dz)\right) \leq \int_\mathcal{Z} \varphi_{\bar{\lambda}, f}(z)(P - P_n)(dz) .$$

By Lemma 4, we have $\bar{\lambda} \in [\zeta^-_{f, P_n}, \zeta^+_{f, P_n}]$. Define the function class $\Phi := \{\varphi_{\lambda, f} : \lambda \in [\zeta^-, \zeta^+], f \in \mathcal{F}\}$. Then, we have

$$R_{\epsilon_\mathcal{B}, 1}(P, f) - R_{\epsilon_\mathcal{B}, 1}(P_n, f) \leq \sup_{\varphi \in \Phi}\left[\int_\mathcal{Z} \varphi(z)(P - P_n)(dz)\right].$$

Since all $f \in \mathcal{F}$ takes values in $[0, M]$, the same holds for all $\varphi \in \Phi$. Therefore, by a standard symmetrization argument [34],

$$R_{\epsilon_\mathcal{B}, 1}(P, f) - R_{\epsilon_\mathcal{B}, 1}(P_n, f) \leq 2\mathfrak{R}_n(\Phi) + M\sqrt{\frac{log(1/\delta)}{2n}}$$

with probability at least $1 - \delta$, where $\mathfrak{R}_n(\Phi) := \mathbb{E}[\sup_{\varphi \in \Phi} \frac{1}{n} \sum_{i=1}^n \sigma_i \varphi(z_i)]$ is the expected Rademacher complexity of $\Phi$. Using the bound of Lemma 5, we get the desired result

$$R_{\epsilon_\mathcal{B}, 1}(P, f) - R_{\epsilon_\mathcal{B}, 1}(P_n, f) \leq \frac{24\mathfrak{C}(\mathcal{F})}{\sqrt{n}} + M\sqrt{\frac{log(\frac{1}{\delta})}{2n}} + \frac{12\sqrt{\pi}}{\sqrt{n}}\Lambda_{\epsilon_\mathcal{B}} \cdot diam(Z) .$$

# D  Proof of Theorem 1

*Proof of Theorem 1.* By Proposition 1, $R_{\epsilon_{\mathcal{B}},1}(P_n, f)$ can be written as

$$
\begin{aligned}
R_{\epsilon_{\mathcal{B}},1}(P_n, f) \quad &= \min_{\lambda \geq 0}\{\lambda\epsilon_{\mathcal{B}} + \mathbb{E}_{P_n}[\varphi_{\lambda,f}(z)]\} = \min_{\lambda \geq 0}\{\lambda\epsilon_{\mathcal{B}} + \mathbb{E}_{P_n}[\varphi_{\lambda,f}(z) - f(z)]\} + \mathbb{E}_{P_n}[f(z)] \\
&= \min_{\lambda \geq 0}\{\lambda\epsilon_{\mathcal{B}} + \psi_{f,P_n}(\lambda)\} + \frac{1}{n}\sum_{i=1}^{n} f(z_i)
\end{aligned}
$$

where the last equality uses the definition of $\psi_{f,P_n}(\lambda)$. Substituting the above equation into Lemma 6 and using inequality (1), we get result (3). To obtain (4), we make use of the following inequality

$$
\min_{\lambda \geq 0}\{\lambda\epsilon_{\mathcal{B}} + \psi_{f,P_n}(\lambda)\} \leq \lambda_{f,P_n}^+ \epsilon_{\mathcal{B}} + \psi_{f,P_n}(\lambda_{f,P_n}^+) = \lambda_{f,P_n}^+ \epsilon_{\mathcal{B}} \quad ,
$$

where the equality follows from the definition of $\lambda_{f,P_n}^+$.

# E  Proof of Corollary 1

*Proof of Corollary 1.* We first verify the assumption conditions in Theorem 1. Assumption 1 is evidently satisfied since $diam(\mathcal{Z}) \leq (2r+1)$. For each $f \in \mathcal{F}$, assumption 2 holds with $M = 1 + \Lambda r$. To verify assumption 3, we can write

$$
\begin{aligned}
f(z') - f(z) \leq \max\{0, yw \cdot x - y'w \cdot x'\} \quad &\leq \max\{0, 2yw \cdot x \mathbb{1}_{(y \neq y')} + \|w\|_2 \|x' - x\|_2\} \\
&\leq \max\{2yw \cdot x, \|w\|_2\} d_{\mathcal{Z}}(z, z')
\end{aligned} \quad ,
$$

where $z = (x, y)$. Then assumption 3 holds with $\lambda_{f,z} = \max\{2yw \cdot x, \|w\|_2\}$. Let $\tilde{\lambda} = \max_i\{2y_i w \cdot x_i, \|w\|_2\}$. By the definition of $\psi_{f,P_n}(\cdot)$, we have $\psi_{f,P_n}(\tilde{\lambda}) = 0$. Since $\lambda_{f,P_n}^+$ is the smallest $\lambda$ which satisfies $\psi_{f,P_n}(\lambda) = 0$, we get $\lambda_{f,P_n}^+ \leq \max_i\{2y_i w \cdot x_i, \|w\|_2\}$.

To evaluate the Dudley entropy integral, we need to estimate the covering numbers $\mathcal{N}(\mathcal{F}, \|\cdot\|_\infty, u/2)$. First observe, for any $f_1, f_2 \in \mathcal{F}$, we have

$$
\|f_1 - f_2\|_\infty = \sup_{x \in \mathcal{X}, y \in \mathcal{Y}} |f_1(x,y) - f_2(x,y)| \leq \sup_{x \in \mathcal{X}, y \in \mathcal{Y}} |yw_1 \cdot x - yw_2 \cdot x| \leq \|w_1 - w_2\|_2 r.
$$

Since $w_1, w_2$ belong to a $\Lambda$-ball in $\mathcal{R}^d$,

$$
\mathcal{N}(\mathcal{F}, \|\cdot\|_\infty, u/2) \leq \left(\frac{6\Lambda r}{u}\right)^d
$$

for $0 < u < 2\Lambda r$, and $\mathcal{N}(\mathcal{F}, \|\cdot\|_\infty, u/2) = 1$ for $u \geq 2\Lambda r$, which gives

$$
\int_0^\infty \sqrt{log\mathcal{N}(\mathcal{F}, \|\cdot\|_\infty, u/2)}du \leq \int_0^{2\Lambda r} \sqrt{d\log(\frac{6\Lambda r}{u})}du \leq 6\Lambda r\sqrt{d} \quad .
$$

Substituting this into expression (4), we get the desired result

$$
R_P(f, \mathcal{B}) \leq \frac{1}{n}\sum_{i=1}^n f(z_i) + \lambda_{f,P_n}^+ \epsilon_{\mathcal{B}} + \frac{144}{\sqrt{n}}\Lambda r\sqrt{d} + \frac{12\sqrt{\pi}}{\sqrt{n}}\Lambda_{\epsilon_{\mathcal{B}}} \cdot (2r+1) + (1 + \Lambda r)\sqrt{\frac{log(\frac{1}{\delta})}{2n}} \quad .
$$

# F  Proof of Corollary 2

The goal of this section is to prove the adversarial expected risk for neural networks. To this end, it is necessary to first establish some properties of the margin operator $\mathcal{M}(v, y) = v_y - \max_{j \neq y} v_j$ and the ramp loss $l_\gamma$.

**Lemma 7.** *For every $j$, $\mathcal{M}(\cdot, j)$ is 2-Lipschitz with respect to $\|\cdot\|_2$.*

*proof.* Let $u, v$ and $y$ be given. If $\mathcal{M}(u, y) \geq \mathcal{M}(v, y)$, denote the index $j$ which satisfies that $\mathcal{M}(v, y) = v_y - v_j$. Then,

$$
\mathcal{M}(u, y) - \mathcal{M}(v, y) = u_y - \max_{i \neq y} u_i - v_y + v_j \leq u_y - u_j - v_y + v_j \leq 2\|u - v\|_\infty \leq 2\|u - v\|_2.
$$

Otherwise, let $j$ be the index satisfying $\mathcal{M}(u, y) = u_y - u_j$, and we obtain

$$
-2\|u - v\|_2 \leq \mathcal{M}(u, y) - \mathcal{M}(v, y).
$$

Therefore, $\mathcal{M}(\cdot, j)$ is 2-Lipschitz with respect to $\|\cdot\|_2$.

**Lemma 8.** *For any* $f \in \mathcal{F}$, *we have* $\lambda^+_{f,P_n} \leq C_4$ *where* $C_4 :=$ $\max_j\{\frac{2}{\gamma}\prod_{i=1}^L \rho_i||A_i||_\sigma, \frac{1}{\gamma}\big(\mathcal{M}(\mathcal{H}_\mathcal{A}(x_j), y_j) + \max \mathcal{H}_\mathcal{A}(x_j) - \min \mathcal{H}_\mathcal{A}(x_j)\big)\}$.

*Proof.* By the definition of $f$, for any $z$ and $z'$, we have

$$
\begin{aligned}
&f(z') - f(z)\\
&= l_\gamma(-\mathcal{M}(\mathcal{H}_\mathcal{A}(x'), y')) - l_\gamma(-\mathcal{M}(\mathcal{H}_\mathcal{A}(x), y))\\
&\leq \max\{0, \frac{1}{\gamma}(\mathcal{M}(\mathcal{H}_\mathcal{A}(x), y) - \mathcal{M}(\mathcal{H}_\mathcal{A}(x'), y'))\}\\
&\leq \max\{0, \frac{1}{\gamma}(\mathcal{M}(\mathcal{H}_\mathcal{A}(x), y') - \mathcal{M}(\mathcal{H}_\mathcal{A}(x'), y')) + \frac{1}{\gamma}(\mathcal{M}(\mathcal{H}_\mathcal{A}(x), y) - \mathcal{M}(\mathcal{H}_\mathcal{A}(x), y'))\}\\
&\leq \max\{0, \frac{2}{\gamma}|\mathcal{H}_\mathcal{A}(x) - \mathcal{H}_\mathcal{A}(x')| + \frac{1}{\gamma}(\mathcal{M}(\mathcal{H}_\mathcal{A}(x), y) - \mathcal{M}(\mathcal{H}_\mathcal{A}(x), y'))\}\\
&\leq \frac{2}{\gamma}\prod_{i=1}^L \rho_i||A_i||_\sigma||x - x'||_2 + \frac{1}{\gamma}\big(\mathcal{M}(\mathcal{H}_\mathcal{A}(x), y) + \max \mathcal{H}_\mathcal{A}(x) - \min \mathcal{H}_\mathcal{A}(x)\big)\mathbb{1}_{y \neq y'}\\
&\leq \max\{\frac{2}{\gamma}\prod_{i=1}^L \rho_i||A_i||_\sigma, \frac{1}{\gamma}\big(\mathcal{M}(\mathcal{H}_\mathcal{A}(x), y) + \max \mathcal{H}_\mathcal{A}(x) - \min \mathcal{H}_\mathcal{A}(x)\big)d_\mathcal{Z}(z, z')
\end{aligned}
$$

where the third inequality uses Lemma 7. Let $C_4 = \max_j\{\frac{2}{\gamma}\prod_{i=1}^L \rho_i||A_i||_\sigma, \frac{1}{\gamma}\big(\mathcal{M}(\mathcal{H}_\mathcal{A}(x_j), y_j) + \max \mathcal{H}_\mathcal{A}(x_j) - \min \mathcal{H}_\mathcal{A}(x_j)\big)\}$. By the definition of $\psi_{f,P_n}(\cdot)$, we have $\psi_{f,P_n}(C_4) = 0$. Therefore, $\lambda^+_{f,P_n} \leq C_4$. $\qquad\square$

**Lemma 9.** *For any two feedforward neural network $\mathcal{H}_\mathcal{A}$ and $\mathcal{H}_{\mathcal{A}'}$ where $\mathcal{A} = (A_1, A_2, \cdots, A_L)$ and $\mathcal{A}' = (A'_1, A'_2, \cdots, A'_L)$, we have the following*

$$
||\mathcal{H}_\mathcal{A}(x) - \mathcal{H}_{\mathcal{A}'}(x)||_2 \leq \prod_{i=1}^L \rho_i s_i B \left(\sum_{j=1}^L \frac{||A_i - A'_i||_\sigma}{s_i}\right).
$$

*Proof.* We prove this by induction. Let $\Delta_i = ||\mathcal{H}^i_\mathcal{A}(x) - \mathcal{H}^i_{\mathcal{A}'}(x)||_2$. First observe

$$
\Delta_1 = ||\sigma_1(A_1 x) - \sigma_1(A'_1 x)||_2 \leq \rho_1||A_1 x - A'_1 x||_2 \leq \rho_1||A_1 - A'_1||_\sigma||x||_2 \leq \rho_1 B||A_1 - A'_1||_\sigma.
$$

For any $i \geq 1$, we have the following

$$
\begin{aligned}
&\Delta_{i+1}\\
&= ||\sigma_{i+1}(A_{i+1}\sigma_i(A_i \cdots \sigma_2(A_2\sigma_1(A_1 x)))) - \sigma_{i+1}(A'_{i+1}\sigma_i(A'_i \cdots \sigma_2(A'_2\sigma_1(A'_1 x))))||_2\\
&\leq ||\sigma_{i+1}(A_{i+1}\sigma_i(A_i \cdots \sigma_2(A_2\sigma_1(A_1 x)))) - \sigma_{i+1}(A'_{i+1}\sigma_i(A_i \cdots \sigma_2(A_2\sigma_1(A_1 x))))||_2 +\\
&\quad ||\sigma_{i+1}(A'_{i+1}\sigma_i(A_i \cdots \sigma_2(A_2\sigma_1(A_1 x)))) - \sigma_{i+1}(A'_{i+1}\sigma_i(A'_i \cdots \sigma_2(A'_2\sigma_1(A'_1 x))))||_2\\
&\leq \rho_{i+1}||A_{i+1} - A'_{i+1}||_\sigma||\sigma_i(A_i \cdots \sigma_2(A_2\sigma_1(A_1 x)))||_2 + \rho_{i+1}s_{i+1}\Delta_i\\
&\leq \rho_{i+1}||A_{i+1} - A'_{i+1}||_\sigma \prod_{j=1}^i \rho_j s_j B + \rho_{i+1}s_{i+1}\Delta_i
\end{aligned}
$$

Therefore, using the induction step, we get the following

$$
\begin{aligned}
&\Delta_{i+1}\\
&\leq \rho_{i+1}||A_{i+1} - A'_{i+1}||_\sigma \prod_{j=1}^i \rho_j s_j B + \rho_{i+1}s_{i+1}\Delta_i\\
&\leq \rho_{i+1}||A_{i+1} - A'_{i+1}||_\sigma \prod_{j=1}^i \rho_j s_j B + \prod_{j=1}^{i+1} \rho_j s_j B \left(\sum_{k=1}^i \frac{||A_k - A'_k||_\sigma}{s_k}\right)\\
&= \prod_{j=1}^{i+1} \rho_j s_j B \left(\sum_{k=1}^{i+1} \frac{||A_k - A'_k||_\sigma}{s_k}\right)
\end{aligned}
$$

$\qquad\square$

We now return to the proof of Corollary 2.

*Proof of Corollary 2.* First we verify the three assumptions. Assumption 1 holds with $diam(\mathcal{Z}) \leq 2r + 1$. Assumption 2 is self-satisfied by the definition of ramp loss with $0 \leq f(z) \leq 1$. And

assumption 3 is guaranteed by Lemma 8 with $\lambda^+_{f,P_n} \leq C_4$. Now we proceed to upper bound the covering number for $\mathcal{F}$. For any $f$ and $f'$,

$$
\begin{aligned}
||f - f'||_\infty &= \sup_z |f(z) - f'(z)| = \sup_z |l_\gamma(-\mathcal{M}(\mathcal{H}_\mathcal{A}(x), y)) - l_\gamma(-\mathcal{M}(\mathcal{H}_{\mathcal{A}'}(x), y))| \\
&\leq \sup_x \frac{2}{\gamma} ||\mathcal{H}_\mathcal{A}(x) - \mathcal{H}_{\mathcal{A}'}(x)||_2 \leq \frac{2}{\gamma} \prod_{i=1}^L \rho_i s_i B \left( \sum_{j=1}^L \frac{||A_j - A'_j||_\sigma}{s_i} \right)
\end{aligned}
\quad,
$$

where the last inequality applies Lemma 9. Since for any matrix $A$, we have $||A||_\sigma \leq ||A||_F$. The above inequality can be written as

$$
||f - f'||_\infty \leq \frac{2}{\gamma} \prod_{i=1}^L \rho_i s_i B \left( \sum_{j=1}^L \frac{||A_j - A'_j||_F}{s_i} \right).
$$

Define $u_j, a_j$ and $\bar{a}$ as

$$
u_j = \frac{s_j u a_j}{\frac{4}{\gamma} \prod_{i=1}^L \rho_i s_i B}, a_j = \frac{1}{\bar{a}} \left( \frac{b_j}{s_j} \right)^{1/2}, \bar{a} = \sum_{j=1}^L \left( \frac{b_j}{s_j} \right)^{1/2}.
$$

So,

$$
\frac{2}{\gamma} \prod_{i=1}^L \rho_i s_i B \left( \sum_{j=1}^L \frac{u_j}{s_j} \right) = \frac{u}{2}.
$$

Then, the covering number $\mathcal{N}(\mathcal{F}, || \cdot ||_\infty, u/2)$ can be bounded by

$$
\begin{aligned}
&\int_0^\infty \sqrt{\log \mathcal{N}(\mathcal{F}, || \cdot ||_\infty, u/2)} du \\
&\leq \int_0^\infty \sqrt{\sum_{i=1}^L \log \mathcal{N}(A_i, || \cdot ||_F, u_i)} du \\
&= \int_0^\infty \sqrt{\sum_{i=1}^L \log \mathcal{N}(\{A_i : ||A_i||_\sigma \leq s_i, ||A_i||_F \leq b_i\}, || \cdot ||_F, u_i)} du \\
&\leq \int_0^\infty \sqrt{\sum_{i=1}^L \log \mathcal{N}(\{A_i : ||A_i||_F \leq b_i\}, || \cdot ||_F, u_i)} du \\
&\leq \int_0^\infty \sum_{i=1}^L \sqrt{\log \mathcal{N}(\{A_i : ||A_i||_F \leq b_i\}, || \cdot ||_F, u_i)} du
\end{aligned}
\quad.
$$

Since $A_i \in \mathbb{R}^{d_i \times d_{i-1}}$, we can regard $A_i$ as a vector in $\mathbb{R}^m$ with $m = d_i \cdot d_{i-1}$ and $|| \cdot ||_F$ as the standard Euclidean distance in $\mathbb{R}^m$. Then the set $\{A_i : ||A_i||_F \leq b_i\}$ forms a $b_i$-ball in $\mathbb{R}^m$, and the covering number for this ball could be upper bounded by

$$
\mathcal{N}(\{A_i : ||A_i||_F \leq b_i\}, || \cdot ||_F, u_i) \leq \left( \frac{3b_i}{u_i} \right)^m \leq \left( \frac{3b_i}{u_i} \right)^{W^2}
$$

for $0 < u_i < b_i$, and $\mathcal{N}(\{A_i : ||A_i||_F \le b_i\}, ||\cdot||_F, u_i) = 1$ for $u_i \ge b_i$. So,

$$\int_0^\infty \sqrt{\log \mathcal{N}(\mathcal{F}, ||\cdot||_\infty, u/2)} du$$

$$\le \sum_{i=1}^L \left( \int_0^\infty \sqrt{\log \mathcal{N}(\{A_i : ||A_i||_F \le b_i\}, ||\cdot||_F, u_i)} du_i \cdot \frac{\frac{4}{\gamma} \prod_{i=1}^L \rho_i s_i B}{s_i a_i} \right)$$

$$\le \sum_{i=1}^L \left( \int_0^{b_i} \sqrt{\log \mathcal{N}(\{A_i : ||A_i||_F \le b_i\}, ||\cdot||_F, u_i)} du_i \cdot \frac{\frac{4}{\gamma} \prod_{i=1}^L \rho_i s_i B}{s_i a_i} \right) ,$$

$$\le \sum_{i=1}^L \frac{\frac{4}{\gamma} \prod_{i=1}^L \rho_i s_i BW}{s_i a_i} \int_0^{b_i} \sqrt{\log \frac{3b_i}{u_i}} du_i$$

$$= \frac{12}{\gamma} \prod_{i=1}^L \rho_i s_i BW \sum_{i=1}^L \frac{b_i}{s_i a_i} \int_0^{\frac{1}{3}} \sqrt{\log \frac{1}{u_i}} du_i$$

$$\le \frac{12}{\gamma} \prod_{i=1}^L \rho_i s_i BW \bar{a}^2$$

where the last inequality uses $\int_0^{\frac{1}{3}} \sqrt{\log \frac{1}{u_i}} du_i = \frac{1}{6}(2\sqrt{\log 3} + 3\sqrt{\pi} erfc(\sqrt{\log 3})) < 1$. Substituting it into Theorem 1, we obtain

$$R_P(f, \mathcal{B}) \le \frac{1}{n} \sum_{i=1}^n f(z_i) + \lambda_{f,P_n}^+ \epsilon_{\mathcal{B}} + \sqrt{\frac{log(1/\delta)}{2n}} + \frac{288}{\gamma\sqrt{n}} \prod_{i=1}^L \rho_i s_i BW \left( \sum_{i=1}^L \left( \frac{b_i}{s_i} \right)^{1/2} \right)^2 +$$

$$\frac{12\sqrt{\pi}}{\sqrt{n}} \Lambda_{\epsilon_{\mathcal{B}}} \cdot (2B + 1)$$

.