[Reviews · NeurIPS 2019]

Reviewer 1



Originality: I find the approach original and interesting, I find that other works have been cited and the section of related work is written clearly and detailed, it gives a nice overview. I think only that it is important to highlight more clearly the differences between [40] and the current work. In particular, it is unclear what is the penalty parameter, and how their method of adversarial training relates to this work - do they optimize a different bound or what quantities do they optimize, and do these quantities show up in the proposed bound? Quality: the work seems complete, and sound for as far as I could check. I could not check all the proofs in detail but I read the work in great detail. The authors are fair in their evaluation of the work, in particular, in the conclusion several avenues for improvement are given and some weaknesses are identified. Clarity: I find the paper well written, and nice to read. The introduction and related work are very well written. The problem statement is clear. At several points in section 4, it is pointed out in text what the reader may expect next, how the proofs will be constructed, etc. which is very helpful. Some examples of bounds are given which is also helpful to understand them better. Significance: I think the paper gives a significant contribution towards providing theoretical guarantees for adversarial learning, which is an important problem to study and improve our understanding of. The proof technique is novel and may also find application elsewhere, even for the analysis of supervised deep neural networks. When reading I took some careful notes which may be helpful to improve the work. Mostly they are language issues, but sometimes also some misunderstandings on my part: - line 36: remove 'we attempt to', turn into 'we fit the adv...' - in line 30: it isn't entirely clear to me why the VC dimension would be important if there are adversaries present, since the bounds based on VC dimension do not apply in this setting, right? maybe it is good to mention that they extend VC theory to deal with adversaries as well. - in line 90: I think you may remove 'during the preparatation of the initial draft of this paper' as it is irrelevant - in line 91: can you explain why it is desireable to obtain regular supervised bounds if $\epsilon_B$ goes to zero? - in line 95: it is unclear what the penalty parameter is - but perhaps the conclusion is enough that 'under condition XYZ their method can only achieve a small amount of robustness' ? btw, does this contradict your earlier statement in sentence 94 'hence certified a level of robustness' ? after having a second look at [40], I think it might be important to highlight a little more the difference between the current paper and their paper, they seem very similar - in line 99: maybe it is not correct that learning systems become unthrustworthy... perhaps it is better to say 'may render predictions made by learning system unthrustworthy' / 'may impact the performance of learning systems significantly'. the latter statement may also be backuped up by literature. - in line 110, you talk about the improved preprint of [26], mainly the downside of that work compared to your work. but maybe it would also be fair to highlight more differences - e.g. 'instead we use a different techniue', I mean, that one bound has a better constant than the other, is not a main concern (it may always be imroved in future work), so it may be more interesting to give a more interesting comparison of the works. if you highlight the difference of their method to your method, I think you can also remove the sentence 'We hope that our method can provide new insight into analysis of the ...' since it will give a new insight via a different method to derrive risk bounds. - in line 121: I would remove 'imperceptible', I mean, if the $\epsilon$ is large enough perhaps its possible to see the perturbations. also, it really depends on the data - image data, etc. for image data it might be imperceptible, for other kinds of data it might be easy to see. - I found definition 2 very confusing. Why expectation? I would just sum over (x,y) from P_n, this is more common notation as far as I know - In line 140, can you explain why the new distribution always lie within the Wasserstein ball? I think the statement is true, it has something to do with optimal transport, but might be good to mention. (AH! You prove this later. This note is too early! --- or maybe should be prepended with a second note, 'which we will show in section 4.x') - line 145, e W_p(P,Q). maybe its good to indicate that (z,z')~M (in the expectation subscript) - line 161, is it correct that the p of the adversary ball N(x) must be the same p as the Wasserstein distance of Lemma 2? Or do these p's not be related? - line 167, 'so, ...' its unclear to me why you only discuss p = 1, does this give the tightest bound? - line 180, if $\lambda$ can depend on f and z, maybe you should indicate it as $\lambda(f,z)$. also, why refer to it as a constant then? - line 185 'Assumption 3 is very straightforward. But it is not easy to use for our proof. For this sake, we 186 give an equivalent expression to Assumption 3 in the following lemma.' replace with: 'Now we give an equivalent expression for Assumption 3 which is easier to use in our proofs.' - line 209, maybe before you can give the lemma, you can say in one or two words how to apply lemma 4 and 5 to get lemma 6. it seems like you use a general rademacher bound on that function class to do that? - in line 224, forgot the word 'bound' after 'generalization' [- lemma 6, theorem 1, line 216, e 4, sometimes you use a dot proceeding diam(Z) and sometimes not. please be consistent - line 231, remove 'our concurrent work' (our!) - line 236, forgot 'equation' before (4) - line 249 remove 'We first define some notations.' - eq 5, please first give the equation and then refer to it. so I would say 'we obtain a risk bound for neural networks: . The bound is in terms of the spectral norm ...' - line 255, 'Although Equation (5) is ....'. - line 258: 'Since our proof technique is different, our approach may provide a different perspective on the generalization of deep neural networks'. - Remark 8: I think this remark is unnecessary, since you will elloborate on it in the Conclusion directly after the remark itself. - line 269, what is the unavoidable dimension dependency? which symbol and which dimension? - I could not understand the last paragraph of the conclusion --- what is eta referring to? I think there might not be enough space to explain the idea clearly here, or you should spend a bit more sentences on the concept... an equation or a reference to an euation would help here. also, before section 5 started I expected you to comment on this issue - how can we optimize these bounds? what do these term represent? it would be very nice if you comment on this earlier.. - generally, I find the conclusion to sound very pessimistic about the work - too much so. Some suggestions. Line 263, remove 'while'. Then, 'The bound may be loose in some cases, since we consider the worst case distribution in the Wasserstein ball in order to avoid computing the transport map. However, for some prob;ems, it may be possible to derrive the transport map and thus get tighter bounds. Furthermore, our bounds may be made tighter by relying on expected Rademacher complexities instead of using covering numbers'. - in the contributions list you said, your bound contains 2 data dependendt terms - but which are these? in the end, I feel like we only talk about one term, $\lambda^+_{f,P_N}$ - in theorem 2, two bounds are given. why is it not interesting to look at the first bound? you never talk about it, which makes me wonder why it is given in theorem 2 at all. interestingly, I believe the two first terms are not exactly the same as the empirical adversarial risk, is that correct? it would be nice if you could say something about this term. could it be optimized? which bound is tighter?

Reviewer 2



It is important to investigate the generalization error bound of adversarial learning. The whole paper is well written and the current analysis improves the related results in Lee & Raginsky [28, 29] .

Reviewer 3



The paper is well written and clearly structured and introduces the necessary background for understanding the full mathematical derivations. The obtained bound may be instantiated (by setting the parameter \epsilon_B =0) to a traditional generalization bound without adversary. Example bounds are provided for SVMs and for multi layered neural networks. The paper recovers previously published bounds but with another derivation, hence opening maybe new tracks for future works or at least providing a new view of these bounds. The paper is quite interesting and focuses on a key issue of deep learning, a theoretical analysis of their behavior under attacks. Yet t is difficult to guess how tight are these bounds, and if they might be useful as is for the design or selection of deep networks. For instance how does the bound in (5) for neural networks without adversary behave with the size if the networks ? Does this bound follow empirical observations (that large overparameterized networks generalize well) ?

[Author Response · NeurIPS 2019]

We appreciate the insightful and constructive comments by all reviewers. All comments will be carefully addressed in
the final version. Below, we provide detailed responses to major concerns.
To Reviewer 1:
> Comparison to [40].
Our paper and [40] differ in the following three aspects. First, [40] considers a Lagrangian relaxation of the local
worst-case risk $R_{\epsilon,p}(P,h) = \sup_{Q:W_p(P,Q))\leq\epsilon} R_Q(h)$ for a fixed penalty parameter $\lambda$, which can be reformulated
as $\sup_Q\{R_Q(h) - \lambda W_p(P,Q)\} = \mathbb{E}_P[\varphi_{\lambda,f}(Z)]$. Then an adversarial training procedure is developed to minimize
$\mathbb{E}_P[\varphi_{\lambda,f}(Z)]$ in order to achieve distributional robustness. However we focus on the original $R_{\epsilon,p}(P,h)$ and expect
to find the optimal $\lambda$ which minimizes $\{\lambda\epsilon_\mathcal{B} + \mathbb{E}_{P_n}[\varphi_{\lambda,f}(Z)]\}$. Specifically, we show that the optimal $\lambda$ falls in
$[\zeta^-_{f,P_n}, \zeta^+_{f,P_n}]$ in Lemma 4. In this way, we are able to obtain a much tighter bound. Moreover, Lemma 4 suggests that
the $\lambda$ in [40] can be selected from the interval $[\zeta^-_{f,P_n}, \zeta^+_{f,P_n}]$. Second, the penalty parameter $\lambda$ must be set to a large
number at the training procedure in [40], and so their method can only deal with gentle adversarial attacks. In contrast,
our bound can deal with arbitrarily large and general adversarial attacks. Finally, the proof techniques in two papers are
different. [40] first relaxes the original local worst-case risk and then provides a generalization bound for the relaxed
problem. In contrast, we prove a uniform bound for the local worst-case risk.
> Discuss how both bounds of theorem 1 compare to each other, ... and how does it relate to the adversarial risk.
The first two terms in both bounds can be deemed as a relaxation of the empirical adversarial risk. They correspond to
the empirical risk and the effect of adversary on empirical risk, respectively. Although the first bound is tighter, it is
hard to optimize because of the inner minimization problem. Therefore, we only discuss the second bound in the paper.
> Update the conclusion to more clearly discuss ..., and how would such an approach differ from what [40] does.
Our bound has two data dependent terms: $1/n\sum_{i=1}^n f(z_i)$ and $\lambda^+_{f,P_n}\epsilon_\mathcal{B}$, corresponding to the empirical risk and the
effect of adversary on empirical risk, respectively. However, in practice, we cannot minimize the sum of the two terms
because $\lambda^+_{f,P_n}$ is computationally intractable. Instead, we consider a heuristic method.
We consider a data-dependent upper bound for $\lambda^+_{f,P_n}$ which is usually easy to obtain. Instead of using the exact $\lambda^+_{f,P_n}$
in the objective function, we consider a regularization parameter $\eta \in [0,1]$ which can be selected via a grid search.
For a fixed $\eta$, we multiply it by the upper bound for $\lambda^+_{f,P_n}$ and use this product as a surrogate of the true $\lambda^+_{f,P_n}$ in the
objective function. Afterward, we minimize this surrogate objective function and obtain the optimal solution for this
specific $\eta$. Each such $\eta$ corresponds to a solution. Finally we choose the best one from these candidates.
The proposed approach is largely different from the training procedure in [40]. In [40], a fixed parameter $\lambda$ is chosen in
advance. Then the training procedure aims to find the optimal $f$ which minimize $\mathbb{E}_{P_n}[\varphi_{\lambda,f}(Z)]$. Their paper focuses
on developing an algorithm for optimizing $\varphi_{\lambda,f}(Z)$. However, our method puts more efforts on finding a good $\lambda$ which
depends on the data and $f$, i.e., $\lambda^+_{f,P_n}$. Once finding $\lambda^+_{f,P_n}$, the optimization would become relatively easier, because
$\psi_{f,P_n}(\lambda)$ in our objective function is 0 when $\lambda$ is set to $\lambda^+_{f,P_n}$.
To Reviewer 2:
> This paper ...better characterize the excess risk bound, ... further provide detail comparisons with related results.
The excess risk bound for adversarial learning can be derived using Theorem 1 and Hoeffding's inequality. Here we
provide the general form of excess risk bound. When applying it to SVMs and deep neural networks, the desired bounds
can be derived. Denote $\bar{f} = \arg\min_{f\in\mathcal{F}} R_{P_n}(f,\mathcal{B})$ and $f^* = \arg\inf_{f\in\mathcal{F}} R_P(f,\mathcal{B})$. The general excess risk bound
can be expressed as $R_P(\bar{f},\mathcal{B}) - R_P(f^*,\mathcal{B}) \leq \lambda^+_{\bar{f},P_n}\epsilon_\mathcal{B} + 24\mathfrak{C}(\mathcal{F})/\sqrt{n} + 12\sqrt{\pi}\Lambda_{\epsilon_\mathcal{B}} diam(Z)/\sqrt{n} + 2M\sqrt{log(2/\delta)/2n}$.
We compare our bounds with related results. For SVMs, our bound is the same as related bounds (e.g., Corollary 4.1,
[34]) except for a dimension dependent factor $\sqrt{d}$, because we use covering number analysis instead of Rademacher
complexity in deriving the bounds. This has been explained in the conclusion. For neural networks, both our bounds
and the bounds in [6, 35] have an explicit dependency on the network size, i.e., $W$. [35] have an additional factor of the
number of layers of networks in their bound. Our work and [35] use spectral norm and Frobenius norm of the weight
matrices, whereas the bound in [6] is given in terms of spectral norm and $(2,1)$ matrix norm. Although the results are
similar in these papers, the proof techniques are different.
To Reviewer 3:
> Provide hints on how the bounds might be useful for Deep Neural Networks design.
Our adversarial risk bounds for deep neural networks might be helpful in the design of neural networks for resisting
adversarial attacks. First, our results show that the adversary would introduce an additional contribution to the empirical
risk. And from the expression for $\lambda^+_{f,P_n}$ in Corollary 2, we can see that this effect could be weakened by the margin
factor $\gamma$. This makes sense since margin can be regarded as a mechanism to defend against adversarial attacks. Therefore,
choosing a relatively large margin value $\gamma$ for which the empirical risk is small can improve the adversarial robustness.
Second, the value $\lambda^+_{f,P_n}$ is closely related to the Lipschitz constant of the function $f$. Our bound indicates that training
the networks with the Lipschitz regularization term (e.g., Virmaux and Scaman, 2018) might be helpful for resisting
adversarial attacks.

[Meta-Review · NeurIPS 2019]

This paper is a contribution that is a step towards theoretical guarantees for adversarial learning. It is timely, well-written with sound theoretical findings. It the authors could provide to empirical evidence of their theoretical findings, this would make the contribution even more compelling.